# Lateral Trajectory Tracking of Self-Driving Vehicles Based on Sliding Mode and Fractional-Order Proportional-Integral-Derivative Control

**Xiqing Zhang** [1,2,*] **, Jin Li** [2,3,*] **, Zhiguang Ma** [2,3] **, Dianmin Chen** [2,3] **and Xiaoxu Zhou** [2,4]

[1] School of Vehicle and Transportation Engineering, Taiyuan University of Science and Technology, Taiyuan 030024, China

[2] Smart Transportation Laboratory in Shanxi Province, Taiyuan 030024, China; mazhiguang2021@163.com (Z.M.); s202112210489@stu.tyust.edu.cn (D.C.); neuzxx@163.com (X.Z.)

[3] School of Mechanical Engineering, Taiyuan University of Science and Technology, Taiyuan 030024, China

[4] Shanxi Intelligent Transportation Research Institute Company Limited, Taiyuan 030024, China

\* Correspondence: xiqingz@tyust.edu.cn or 2011062@tyust.edu.cn (X.Z.); lijin8869@163.com (J.L.)

**Abstract:** The tracking accuracy and vehicle stability of self-driving trajectory tracking are particularly important. Due to the influence of high-frequency oscillation near the sliding mode surface and the modeling error of the single-point preview model itself when using sliding mode control (SMC) for the trajectory tracking lateral control of self-driving vehicles, the desired tracking effect of self-driving vehicles cannot be achieved. To address this problem, a combination of sliding mode control and fractional-order proportional-integral-derivative control (FOPID) is proposed for the application of a trajectory tracking lateral controller. In addition, in order to compare with the trajectory tracking controller built using the single-point preview model, 12 real drivers with different levels of proficiency were selected for operational data collection and comparison. The simulation results and hardware-in-the-loop results show that the designed SMC + FOPID controller has high tracking accuracy based on vehicle stability. The trajectory accuracy based on SMC + FOPID outperforms the real driver data, SMC controller, PID controller, and model prediction controller.

**Keywords:** self-driving vehicles; trajectory tracking control; lateral control; sliding mode control; fractional-order proportional-integral-derivative control





## 1. Introduction

The emergence of self-driving vehicles makes it a popular research object as traffic congestion rises and traffic accidents occur increasingly frequently [1–4]. Trajectory tracking control is committed to ensuring tracking accuracy and vehicle stability, and according to the predetermined trajectory of the vehicle, it can change the vehicle movement in real time, which is considered one of the key research techniques [5,6]. Lateral motion control is an important part of trajectory tracking for autonomous vehicles, and is a key part of ensuring vehicle safety and stability [7,8].

With increasing attention to the research of autonomous driving control technology, certain results have been achieved for research into the lateral control of self-driving vehicles. Fraikin et al. proposed a hybrid vehicle model that combines a vehicle monorail model with a long short-term memory neural network, which not only reduces model computation time but also enables more accurate long-term prediction of vehicle lateral dynamics [9]. Guo et al. proposed a double envelope path tracking method using a model prediction algorithm with variable sampling time and variable prediction step, which can effectively deal with the modeling error and improve the path tracking accuracy [10]. Lin et al. combined a linear time-varying model prediction method with a sliding mode control method to establish a combined control framework to improve the reliability of lateral and longitudinal control, and used direct transverse sway moment control to ensure the good

transverse sway stability of the vehicle in trajectory tracking [11]. Xu et al. presented a predictive steering control algorithm for closed-loop system analysis and experimentally verified that this algorithm is suitable for the accurate, smooth, and computationally inexpensive path tracking of self-driving vehicles [12]. Zhang et al. used a trajectory planning state hierarchy approach to design a controller based on vehicle kinematics to facilitate curve tracking [13]. Lin et al. designed a preview controller based on a simulated annealing algorithm to optimize the preview distance, which can use the road curvature as preview information for feedforward control and feedback control, and it was verified that this method can improve the robustness of the controller under different working conditions [14]. Mata et al. introduced a constant nominal longitudinal speed calculation method into the robustness control algorithm based on a linear time-invariant monorail model, which guarantees vehicle comfort criteria over a wide range of speeds [15]. Yang et al. presented a feedforward + predictive linear quadratic regulator lateral control method based on the vehicle dynamics error model, and it was verified that the control method is suitable for the lateral tracking of vehicles under complex operating conditions [16]. Chen et al. proposed a lateral motion model prediction controller based on an online adaptive method of tire parameters, which effectively improves the robustness of the controller [17]. Awad et al. combined fuzzy logic switching rules with model predictive control algorithms to develop a control strategy that improves trajectory tracking accuracy in different driving scenarios [18].

Various advanced control methods have been explored by researchers, and the sliding mode control (SMC) method is a control method that has been studied by many scholars and has been applied in many fields [19–26]. Since the SMC control method has the advantages of being a more robust, fast system response, and does not require an accurate model, many scholars have applied it to transverse controllers [27,28]. However, during the sliding mode control, the high-frequency transitions generated near the sliding surface and the steady-state errors caused by the model modeling of the single-point preview model can lead to a high-frequency jitter vibration phenomenon, which can affect the accuracy of the control and cause the system to vibrate or cause vehicle instability in severe cases [29]. Yin et al. proposed a self-tuning SMC method based on lateral bias to reduce the sliding mode controller jitter by introducing a gain value of the switching function that can follow the system tracking error and the sliding mode surface [30]. Xia et al. designed a road automatic identification system delayed trajectory tracking sliding mode controller, which effectively improves the safety and stability of trajectory tracking vehicles [31].

Inspired by the literature review, in order to address the robustness problem of using transverse motion sliding mode control, to reduce the steady state error caused by modeling the single-point preview model, and to reduce the high-frequency vibration jitter phenomenon, this paper combines SMC with FOPID. The main contributions of this paper are as follows:

1.  Based on the two-degree-of-freedom dynamics model and single-point preview model, an SMC + FOPID trajectory tracking lateral motion controller is proposed.
2.  Integral order and derivative order can be freely adjusted from 0 to 2 in the FOPID controller, which will extend the lag phase angle for fractional order integrals and the overtravel phase angle for derivatives from 0°~90° to 0°~180°. This allows for more comprehensive parameter tuning, as well as a memory function for the integral and derivative terms that allows the system to achieve better control. The FOPID controller plays a role in compensating the tracking error to the SMC controller and enhances the operational flexibility of the control system.
3.  Based on the hardware device, data acquisition of realistic driver operations with different driving experiences under preset road conditions is achieved.
4.  Simulation and hardware-in-the-loop comparison experiments yielded that the overall control performance of the designed controller outperforms that of the selected driver data, SMC controller, PID controller, and model prediction controller.

This paper is organized as follows. In Section 2, a two-degree-of-freedom dynamics model and a single-point preview model are developed. The SMC + FOPID trajectory tracking lateral motion controller is designed in Section 3. Section 4 collects operational data from realistic drivers with different driving experiences. In Section 5, the effectiveness of the designed controller is verified through simulation and hardware-in-the-loop. Conclusions are drawn in Section 6.

## 2. Control System Model for Lateral Motion

### 2.1. Vehicle Dynamics Model

In order to ensure the predictability of the trajectory tracking control, the controller is designed based on a two-degree-of-freedom vehicle dynamics model [32,33], and the following assumptions are made for the vehicle: (1) Ignore the effects of the vehicle due to vertical, pitch, and sideways motion. (2) Ignore the difference of side deflection characteristics caused by left and right tire loads, and consider the vehicle to have the same left and right wheel turning angles when driving. (3) Neglect the influence of aerodynamics. These influences affect the accuracy of trajectory tracking when real self-driving vehicles are traveling and are not considered in this paper. On this basis, only lateral and transverse motion are considered, and the longitudinal speed is assumed to remain constant. The established two-degree-of-freedom vehicle dynamics model is shown in Figure 1.

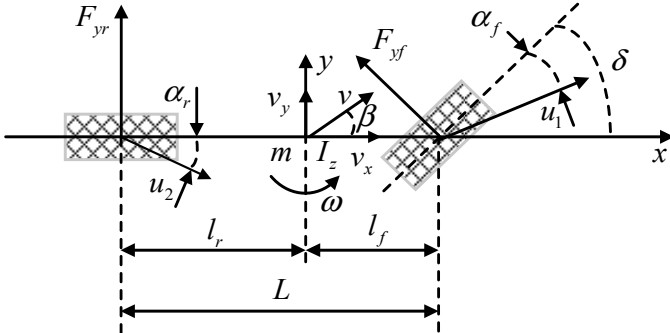

**Figure 1.** The two-degree-of-freedom vehicle dynamics model.

According to the two-degree-of-freedom vehicle dynamics model [34], the longitudinal vehicle speed is assumed to be constant, and the equation is formulated as follows:

$$\begin{cases} \dot{\beta} = \left( \frac{l_f C_f - l_r C_r}{m v_x} - 1 \right) \omega + \frac{C_f + C_r}{m v_x} \beta - \frac{C_f}{m v_x} \delta \\ \dot{\omega} = \frac{l_f{}^2 C_f + l_r{}^2 C_r}{I_z v_x} \omega + \frac{l_f C_f - l_r C_r}{I_z} \beta - \frac{l_f C_f}{I_z} \delta \end{cases} \tag{1}$$

where $\dot{\omega}$, $\dot{\beta}$ are the derivative of the yaw angle rate and the derivative of the sideslip angle of the vehicle, respectively; $\omega$, $\beta$ are the yaw rate of the vehicle and the sideslip angle, respectively; $l_f$ and $l_r$ are the distances from the vehicle center of mass to the front and rear axles, respectively; $C_f$ and $C_r$ are the vehicle cornering stiffness of the front tire and rear tire, respectively; $\delta$ is the front wheel angle of the vehicle; $I_z$ is the vehicle rotational inertia around the z axis; $v_y$ and $v_x$ are the vehicle lateral speed and vehicle longitudinal speed, respectively; and $m$ is the mass of the whole vehicle.

Assuming that the vehicle is in steady-state motion, $\omega$ is a constant value, and then it is deduced that $\dot{\beta}$ and $\dot{\omega}$ are both 0. Combining Equation (1), the steady-state gain is defined as the ratio of the vehicle's yaw rate compared to the front wheel angle of the vehicle, which can be expressed as:

$$R_w = \frac{\omega}{\delta} = \frac{v_x}{L(1 + K v_x^2)} \tag{2}$$

$$K = \frac{m}{L^2}\left(\frac{l_f}{C_r} - \frac{l_r}{C_f}\right) \tag{3}$$

where $L$ is the distance from the front axis to the rear axis, and $K$ is the stability factor.

### 2.2. Single-Point Preview Model

In the real driving process, when the vehicle deviates from the predetermined trajectory, within a certain period of time, the driver will adjust the front wheel angle according to the view of the road ahead, so as to reduce the tracking error of the vehicle. The single-point preview model is used to simulate the real driver model for the lateral control of the trajectory tracking [35,36].

Assuming that the yaw rate of vehicle tracking is constant, the lateral vehicle speed is much smaller than the longitudinal vehicle speed, and the vehicle performs a uniform circular motion along the tangent direction of the target trajectory, a single-point preview model is established, as shown in Figure 2.

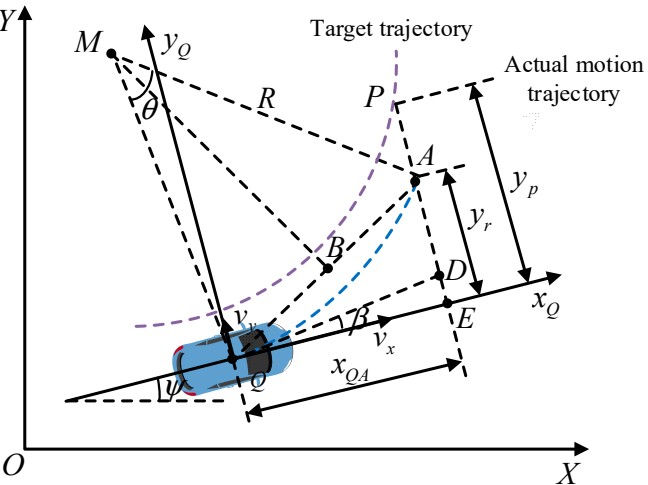

**Figure 2.** The single-point preview model.

In Figure 2, the point $M$ is the center of the circle of circular motion; the point $Q$ is the position of the center of mass at the current moment; the arc $QA$ is the actual trajectory of the vehicle; define $t_p$ as the preview time; the point $A$ is the predicted position of the vehicle center of mass point after the vehicle passes through $t_p$; the point $P$ is the desired target matching point after the vehicle passes through $t_p$; $x_{QA}$ is the longitudinal displacement of the vehicle after passing through $t_p$; $y_p$ is the lateral deviation of the vehicle from the matching point; and $y_r$ is the lateral displacement of the vehicle after $t_p$.

From the analysis of Figure 1:

$$\theta = \omega t_p \tag{4}$$

$$x_{QA} = v_x t_p \tag{5}$$

$$y_r = \tan\left(\frac{\theta}{2} + \beta\right) x_{QA} \tag{6}$$

In the desired tracking process, the vehicle is approximately equal between $y_r$ and $y_p$ at this moment after the preview time; however, the ideal yaw rate can be expressed as:

$$\omega_d = 2\left[\arctan\left(\frac{y_p}{v_x t_p}\right) - \beta\right] t_p^{-1} \tag{7}$$

## 3. Lateral Control Strategy

Based on the sliding mode controller for trajectory tracking, the fractional-order proportional-integral-derivative controller is set as the compensating steering, and the overall control strategy of the controller, combining sliding mode control and fractional-order proportional-integral-derivative control, is shown in Figure 3.

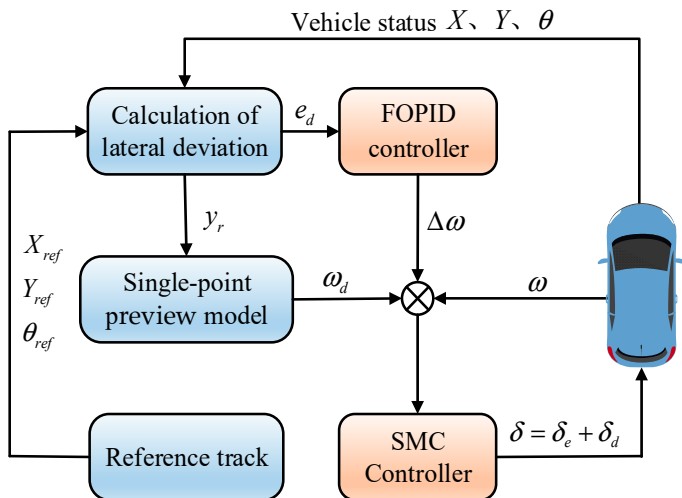

**Figure 3.** Lateral control strategy based on SMC + FOPID.

### 3.1. Design of Sliding Mode Controller

To compensate for the lack of accuracy of the two-degree-of-freedom vehicle model compared to the actual vehicle, a sliding mode controller is built to correct for the modeled parameter perturbations and external uncertainty perturbations.

Equation (1) is reduced to the form of a state space equation as follows:

$$\dot{X}(t) = AX(t) + BU(t) \tag{8}$$

where: $A = \begin{bmatrix} \frac{l_f^2 C_f + l_r^2 C_r}{I_z v_x} & \frac{l_f C_f - l_r C_r}{I_z} \\ \frac{l_f C_f - l_r C_r}{m v_x^2} - 1 & \frac{C_f + C_r}{m v_x} \end{bmatrix}$; $B = \begin{bmatrix} -\frac{l_f C_f}{I_z} \\ -\frac{C_f}{m v_x} \end{bmatrix}$; $U(t) = \delta$

The sliding mode controller adjusts the tracking error between the actual yaw rate and the ideal yaw rate using the sliding mode surface as the reference, as follows:

$$e = \omega - \omega_d \tag{9}$$

The form of the sliding mode surface is shown below:

$$s = \dot{e} + \eta \int_0^t e(\tau) d\tau \tag{10}$$

where $\eta$ is the gain of the sliding mode controller, $\eta > 0$.

The control quantity of the slide mode controller consists of the equivalent control quantity and the switching control quantity, and the final output of the front wheel angle of the vehicle is shown as follows:

$$\delta = \delta_e + \delta_d \tag{11}$$

where $\delta_e$ is the equivalent control quantity and $\delta_d$ is the switching control quantity.

Neglecting the parameter perturbations and uncertain external disturbances of the system, the equivalent control quantity at $\dot{s} = 0$ and $\dot{\omega}_d = 0$ can be derived as:

$$\delta_e = \frac{I_z}{l_f C_f} \left[ \frac{l_f{}^2 C_f + l_r{}^2 C_r}{I_z v_x} \omega + \frac{l_f C_f - l_r C_r}{I_z} \beta + \eta(\omega - \omega_d) \right] \tag{12}$$

In order to correct the disturbance of uncertainties and make the control system stabilize under the slip mode surface, the switching function used is the sigmoid function, which has the form shown below:

$$\delta_d = C_1 \text{sig}(s) = C_1 \left( \frac{2}{1 + e^{-s}} - 1 \right) \tag{13}$$

The stability of the system is proved by using a Lyapunov function for this control system, setting $V = s^2/2$, and deriving it to obtain:

$$\dot{V} = s\dot{s} = s \left[ \frac{l_f^2 C_f + l_r^2 C_r}{I_z v_x} \omega + \frac{l_f C_f - l_r C_r}{I_z} \beta - \frac{l_f C_f}{I_z} (\delta + d(t)) - \dot{\omega}_d + \eta(\omega - \omega_d) \right] \tag{14}$$

Then, Equations (11) and (13) are brought into Equation (14) to obtain:

$$\dot{V} = -s \left[ \frac{l_f C_f}{I_z} (C_1 \text{sig}(s) + d_1(t)) \right] = -sC\text{sig}(s) - d(t)s \tag{15}$$

where $d(t)$ indicates the presence of modeled parameter perturbations or the effect of external perturbations in the system.

The sig function can be regarded as the product of $\text{sgn}(s)$ and $\text{sig}(s)$ because it is an odd function and accepts values in the range of [–1, 1], expressed as follows:

$$\text{sig}(s) = |\text{sig}(s)| \text{sgn}(s) \tag{16}$$

$$\text{sgn}(s) = \begin{cases} 1 & if\ s \geq 0 \\ -1 & if\ s < 0 \end{cases} \tag{17}$$

According to Lyapunov stability theory, the control system is asymptotically stable when $V \leq 0$, which is simplified to:

$$C = \frac{\text{MAX}(|d(t)|) + \varsigma}{|\text{sig}(s)|} \quad \varsigma > 0 \tag{18}$$

This further leads to:

$$\dot{V} = -C|s| - d(t)s \leqslant -\varsigma \frac{|s|}{|\text{sig}(s)|} \leqslant 0 \tag{19}$$

At this point the vehicle control system is progressively stabilized.

### 3.2. Design of Fractional-Order Proportional-Integral-Derivative Controller

The FOPID controller is introduced on the basis of the conventional SMC controller in order to correct the tracking error and increase the operational flexibility of the control system. The control law is depicted in Figure 4. The specific form of the FOPID controller is shown below:

$$\Delta\omega = K_p e(t) + K_i s^{-\chi} \int_0^t e(t)dt + K_d s^{\gamma} \frac{de(t)}{dt} \tag{20}$$

where $K_p$, $K_i$, $K_d$ are the proportional, integral, and derivative gains, respectively; $\chi$ is the integral order; $\gamma$ is the derivative order; $s^{-\chi}$ and $s^{\gamma}$ denote the integral of order $\chi$ and derivative of order $\gamma$ with respect to the system error $e(t)$, respectively; and $\Delta\omega$ are the ideal yaw rate compensation amounts.

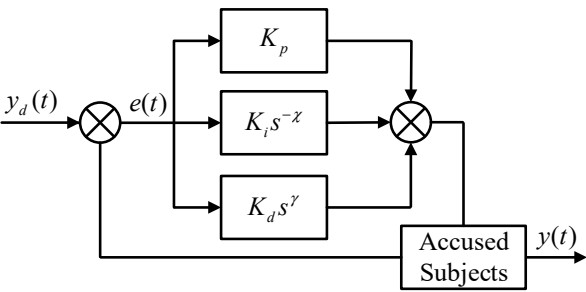

**Figure 4.** FOPID control law [37,38].

$\chi$ and $\gamma$ can be freely adjusted from 0 to 2 in the FOPID controller, which will extend the lag phase angle for fractional order integrals and the overtravel phase angle for derivatives from 0°~90° to 0°~180°. This makes the parameter tuning more comprehensive, while the memory function of the integral and derivative terms is that the system can achieve more optimal control.

## 4. Driver Operation Data Collection and Analysis

Twelve real drivers with different driving experience levels were selected to familiarize themselves with the driver operation platform for a period of time, and then they were allowed to perform lane changing conditions through the driver operation platform. Finally, the steering wheel manipulation data and vehicle status data were collected. The driver operation platform is shown in Figure 5.

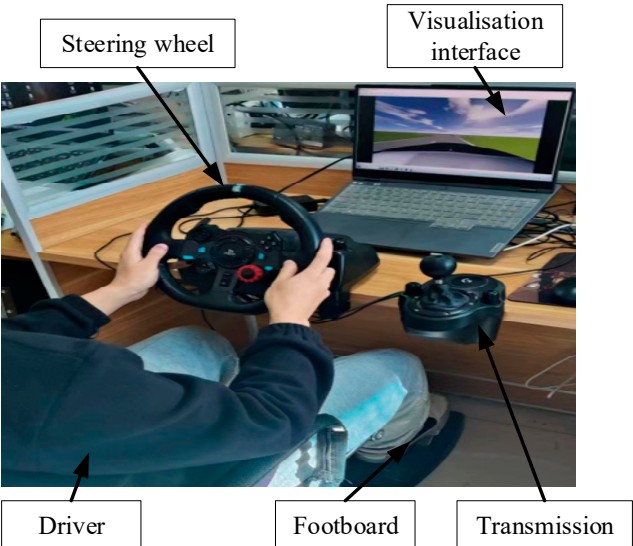

**Figure 5.** Driver operating platform.

Based on an extensive evaluation of vehicle position change and steering control, the best of each parameter was determined to be the skilled driver. Figure 6 displays the vehicle state parameters for the other 11 drivers, the skilled driver, and the average outcomes for all drivers.

As shown in Figure 6a, the skilled driver and the calculated average of all drivers have more flexible control over the vehicle position in the lane change condition compared to the other drivers. Figure 6b–d shows that the skilled driver can maintain a better steering wheel angle, yaw angle, and yaw angle rate, and can enter a more stable state with steering control after lane change; the steering control is also better than that of other drivers and the average of all drivers.

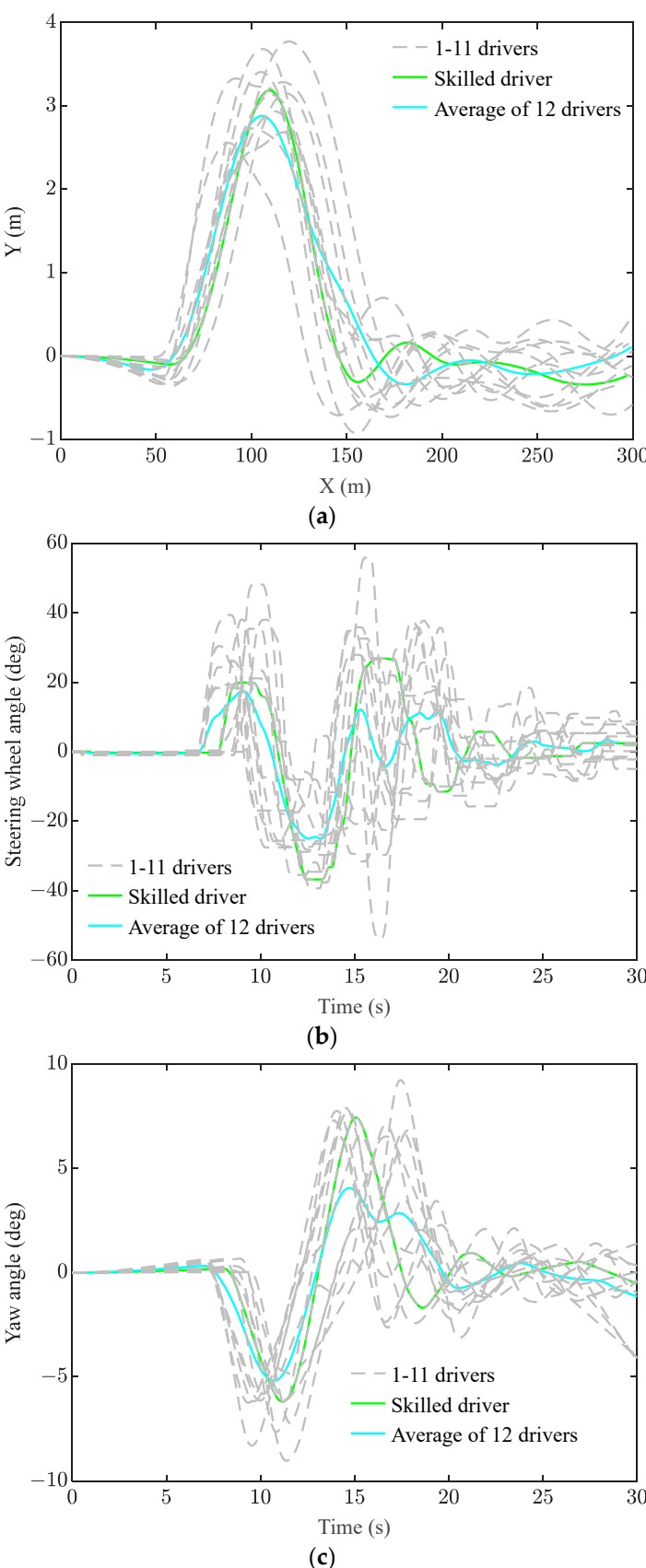

**Figure 6.** *Cont.*

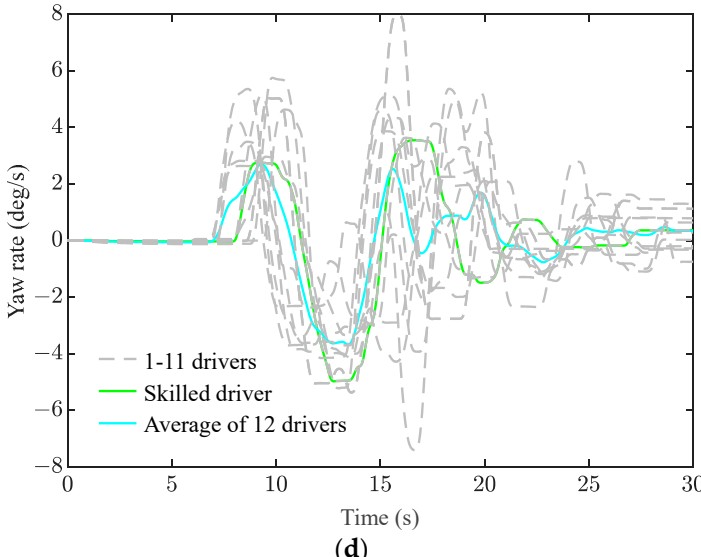

**(d)**

**Figure 6.** Driver operating data. (**a**) Lateral position, (**b**) steering wheel angle, (**c**) yaw angle, and (**d**) yaw rate.

## 5. Simulation and Hardware-in-the-Loop Test

In order to verify the effectiveness of the designed SMC + FOPID lateral motion controller in path tracking, firstly, a joint simulation test based on Matlab/Simulink 2020a and CarSim 2019 software is carried out to compare and analyze with the traditional SMC controller and PID controller under different speed working conditions to verify the control effect of the SMC + FOPID lateral motion controller. Then, hardware-in-the-loop tests are conducted based on the real steering information provided by Logitech G29 driving simulator and the scene provided by PreScan 8.5 to verify the feasibility of the SMC + FOPID lateral motion controller under real steering signals.

### 5.1. Simulation Verification under Different Speed Conditions

The vehicle dynamics model is built in CarSim and the trajectory tracking controller is built in Simulink. The reference trajectory is set up to be the ISO 3888-1:2018 standard double-shift lane change condition with the road central lane as the ideal path [39], and the speed conditions are 30 km/h, 60 km/h, and 90 km/h, with a road surface coefficient of 0.8. The preview time is 0.4 s at 30 km/h, 0.5 s at 60 km/h, and 0.6 s at 90 km/h. The integration order and derivative order of FOPID are both 2. The simulated vehicle parameters are shown in Table 1.

**Table 1.** The simulated vehicle parameters.

| Parameters | Units | Values |
|---|---|---|
| Vehicle weight ($m$) | kg | 1273 |
| Moment of inertia about Z axis ($I_z$) | kg·m$^2$ | 1523 |
| Distance from centroid to front axle ($l_f$) | m | 1.016 |
| Distance from centroid to rear axle ($l_r$) | m | 1.562 |
| Cornering stiffness of the front tire ($C_f$) | N·rad$^{-1}$ | 108,861 |
| Cornering stiffness of the rear tire ($C_r$) | N·rad$^{-1}$ | 108,861 |
| Steering system conventional ratio ($i_{sw}$) | - | 17.6 |

In order to verify the trajectory tracking effect based on the SMC + FOPID controller, it is compared with the traditional SMC controller and the traditional PID controller, respectively. The trajectory tracking effects of the vehicle at 30 km/h, 60 km/h, and 90 km/h are shown in Figure 7, Figure 8, and Figure 9, respectively.

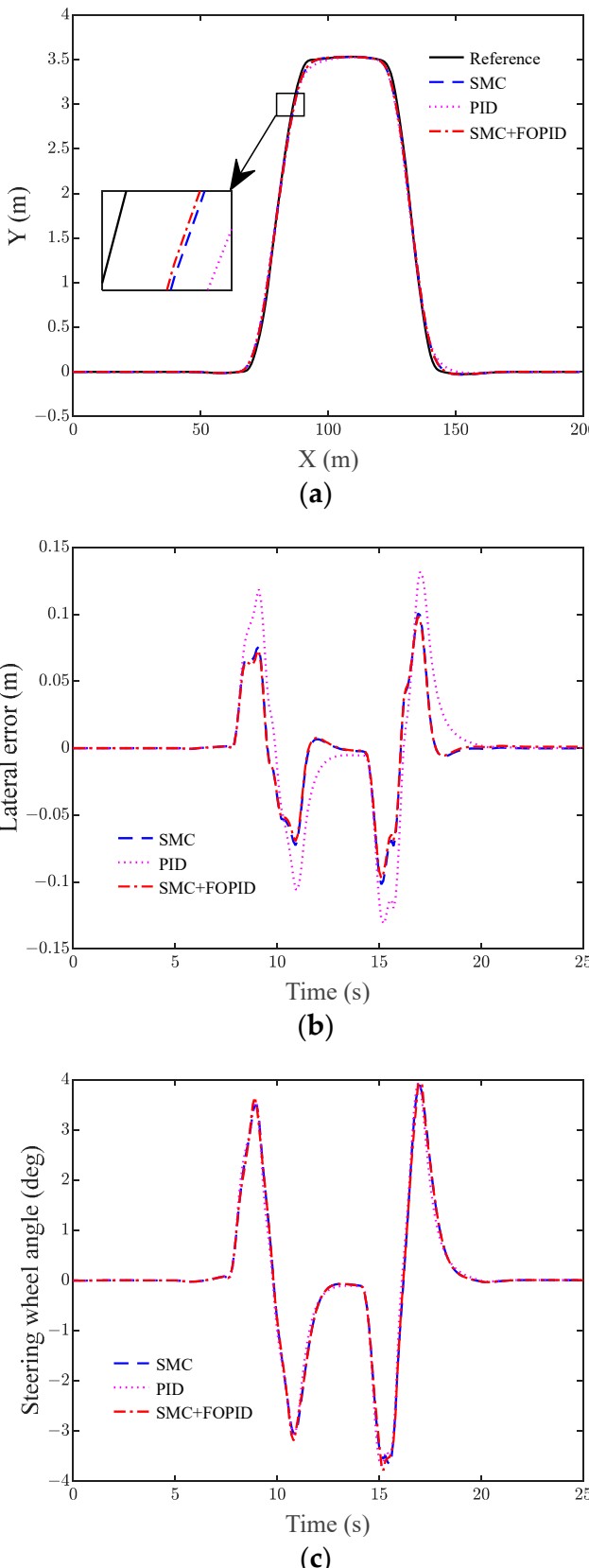

**Figure 7.** Simulated vehicle states for 30 km/h trajectory tracking. (**a**) Lateral position, (**b**) lateral error, and (**c**) steering wheel angle.

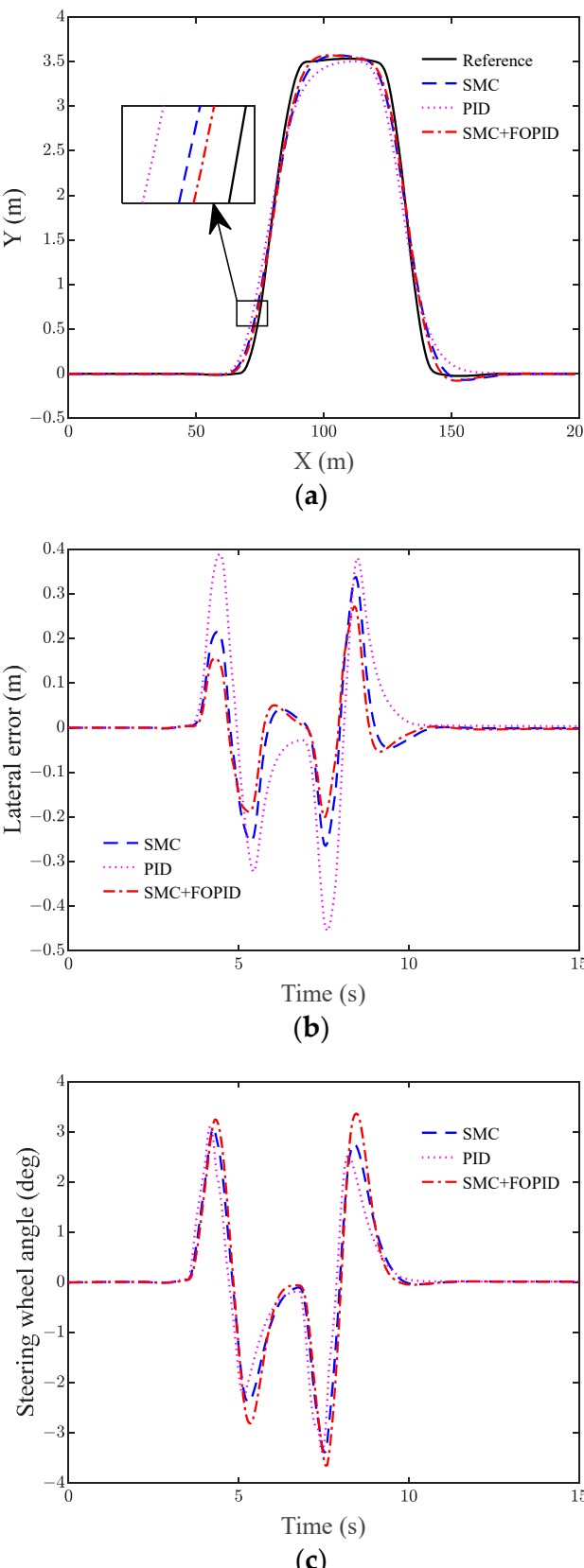

**Figure 8.** Simulated vehicle states for 60 km/h trajectory tracking. (**a**) Lateral position, (**b**) lateral error, and (**c**) steering wheel angle.

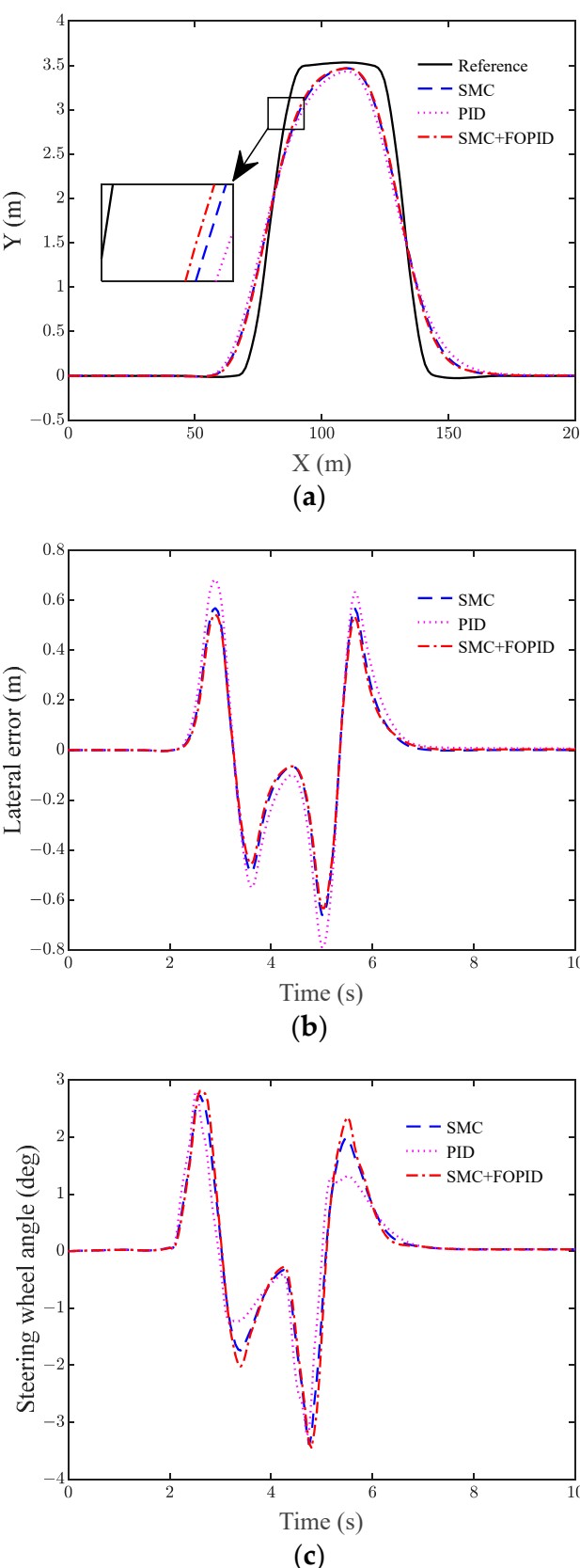

**Figure 9.** Simulated vehicle states for 90 km/h trajectory tracking. (**a**) Lateral position, (**b**) lateral error, and (**c**) steering wheel angle.

Calculations based on Figures 7–9 yielded error data, as shown in Table 2.

**Table 2.** Simulated vehicle tracking error under the double-shifted lane condition.

| Velocity (km/h) | Root Mean Square of Lateral Error (m) | | | Max Lateral Error (m) | | |
|---|---|---|---|---|---|---|
| | SMC | PID | SMC + FOPID | SMC | PID | SMC + FOPID |
| 30 | 0.031 | 0.044 | 0.029 | 0.101 | 0.132 | 0.098 |
| 60 | 0.094 | 0.142 | 0.072 | 0.339 | 0.389 | 0.273 |
| 90 | 0.220 | 0.263 | 0.207 | 0.570 | 0.684 | 0.544 |

As known from Table 2, at 30 km/h, the root mean square value of the lateral error and the maximum lateral error of the SMC + FOPID controller can be controlled within 0.029 m and 0.098 m, respectively. Neither SMC nor PID control is as effective as SMC + FOPID. The root mean square value of the lateral error and the maximum lateral error of SMC are 0.031 m and 0.101 m, respectively. The root mean square value and maximum lateral error of PID are 0.044 m and 0.132 m, respectively.

According to Table 2, it is known that at 60 km/h, the root mean square value of the lateral error and the maximum value of the lateral error are reduced for the SMC + FOPID controller compared to the SMC controller and the PID controller. The root mean square value of the lateral error for SMC + FOPID is lower by 21.7% and 49.3%, respectively, and the maximum value of the lateral error for SMC + FOPID is lower by 19.5% and 29.8%, respectively.

As the vehicle speed becomes faster and the directional sensitivity becomes higher, the control effect of the controller also gradually becomes worse. As we know from Figure 9 and Table 2, at 90 km/h, the root mean square value of the lateral error and the maximum value of the lateral error of the SMC controller reach 0.220 m and 0.570 m, respectively, while the root mean square value of the lateral error and the maximum value of the lateral error of the PID controller reach 0.263 m and 0.684 m, respectively. The root mean square value of the lateral error and the maximum value of the lateral error of the SMC + FOPID controller are 0.207 m and 0.544 m, respectively, and the tracking accuracy of the control of SMC + FOPID is higher than that of SMC and PID. In addition, during the tracking trajectory, the steering wheel angle of the SMC + FOPID controller is kept within 4° for all three speed conditions, without overshooting and within the stable range.

In summary, the SMC + FOPID controller has a stronger trajectory tracking capability than the traditional SMC controller and PID controller, while ensuring vehicle stability.

*5.2. Hardware-in-the-Loop Test*

In order to further verify the trajectory tracking effect of the above SMC + FOPID controller in a near-real state, a hardware-in-the-loop testbed is established, as shown in Figure 10, which is built based on the Logitech G29 driving simulator suite, Prescan 8.5, CarSim 2019, and Matlab/Simulink 2020a. The computation is provided by the Nuvo-8108GC on-board GPU platform.

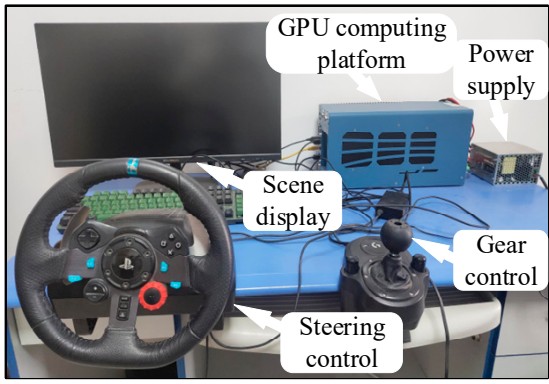

**Figure 10.** Hardware testing platform.

The Logitech G29 driving simulator kit is connected to the computer side via a USB interface, and the vehicle dynamics model built by CarSim is passed to the control module built by Matlab, and then combined with the scenario built by PreScan for real-time automatic steering to form a closed-loop control. The hardware test strategy of the automatic driving control is shown in Figure 11.

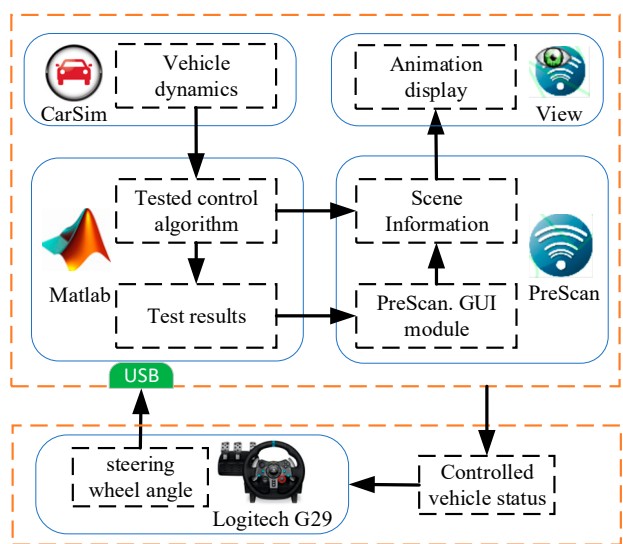

**Figure 11.** The hardware-in-the-loop testing strategy.

### 5.2.1. Double-Shifted Lane Condition Test

The ISO 3888-1:2018 standard double-shifted lane with the road central lane as the ideal path is used as the reference trajectory. The road surface coefficient is 0.8, 36 km/h is used as the speed condition of the experiment, the preview time is 0.4 s, the integral order and the derivative order of the FOPID are both 2, and the vehicle parameters are shown in Table 1.

Model predictive control (MPC) is one of the important methods to study trajectory tracking, and MPC is utilized to implement trajectory tracking and compare it with SMC + FOPID in this paper. Figure 12 demonstrates the hardware-in-the-loop test results for the skilled driver, PID controller, SMC controller, SMC+FPID controller, and MPC controller under the double-shifted lane condition.

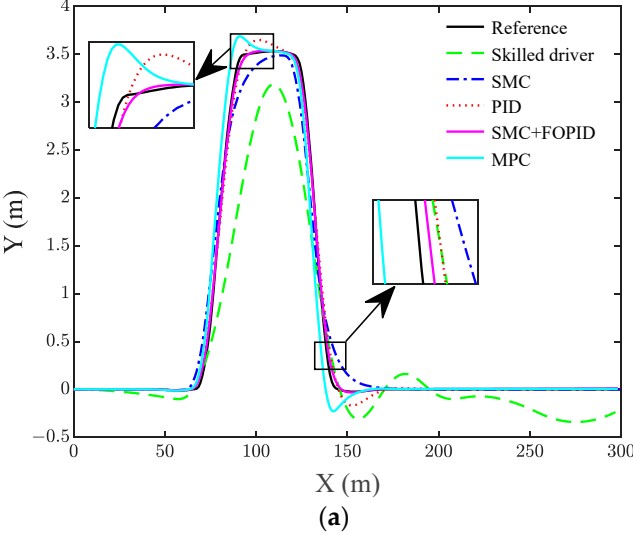

(a)

**Figure 12.** *Cont.*

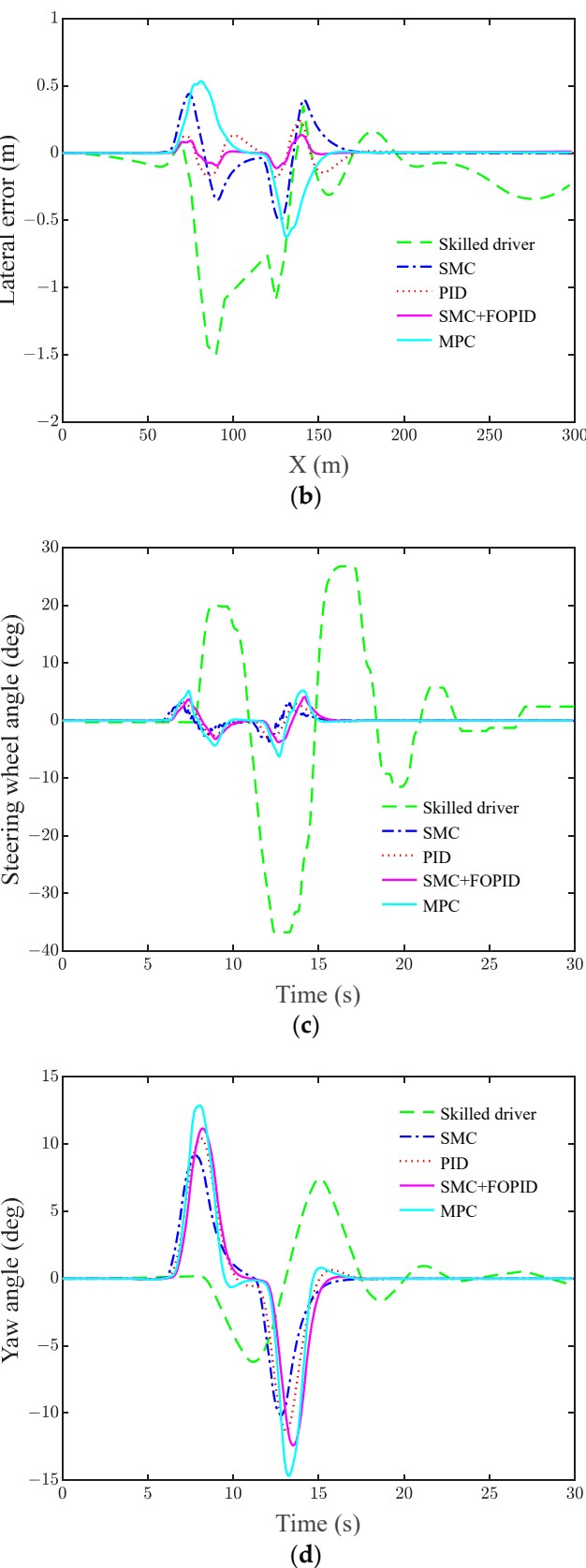

**Figure 12.** *Cont.*

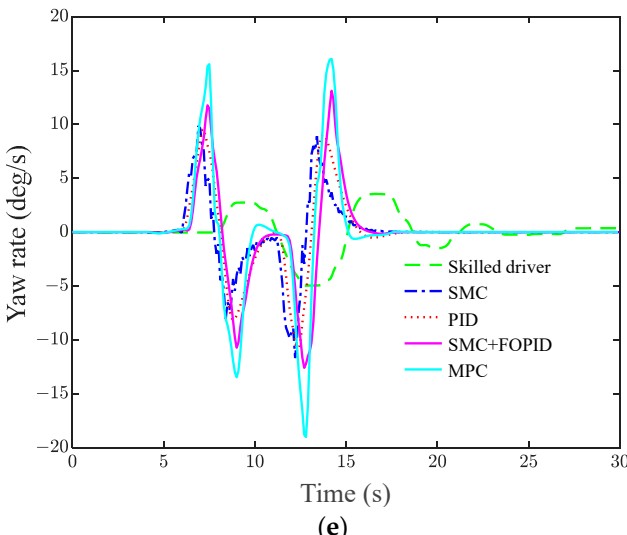

(**e**)

**Figure 12.** Hardware-in-the-loop test results for different control schemes under the double-shifted lane condition. (**a**) Lateral position, (**b**) lateral error, (**c**) steering wheel angle, (**d**) yaw angle, and (**e**) yaw rate.

Table 3 is obtained from the data points of Figure 12 and represents the root mean square value of the lateral error and the maximum value of the lateral error for controller trajectory tracking.

**Table 3.** Hardware-in-the-loop tracking error under the double-shifted lane condition.

| Controller Type | Root Mean Square of Lateral Error (m) | Max Lateral Error (m) |
|---|---|---|
| Skilled drivers | 0.136 | 1.501 |
| SMC | 0.202 | 0.503 |
| PID | 0.099 | 0.239 |
| SMC + FOPID | 0.016 | 0.139 |
| MPC | 0.074 | 0.627 |

Combined with Table 3, it can be seen that the root mean square value of the lateral error for the skilled driver, SMC, PID, and MPC are 0.136 m, 0.202 m, 0.099 m, and 0.074 m, respectively. The root mean square value of the lateral error for SMC + FOPID is 0.016 m, which is less than that for the skilled driver, SMC, PID, and MPC. The maximum values of the lateral error for the skilled driver, SMC, PID, and MPC are 1.501 m, 0.503 m, 0.239 m, and 0.627 m, respectively. The root mean square value of the lateral error for SMC + FOPID is 0.139 m, which is also smaller than that of the skilled driver, SMC, PID, and MPC. In addition, from Figure 12c,d, the maximum steering wheel angle and maximum yaw angle of SMC + FOPID are 4.1° and 10.1°, respectively. From Figure 12e, it can be seen that there is no jitter vibration in the yaw rate curve, and the vehicle maintains a stable state.

In summary, under the dual shift lane changing condition, the trajectory of SMC + FOPID controller is closer to the reference trajectory than the trajectories of Skilled drivers, SMC, PID, and MPC on the basis of ensuring vehicle stability.

5.2.2. U-Shaped Road Test

The central lane of the road is the ideal path of the U-shaped line as a reference trajectory. The road surface coefficient is 0.8, 36 km/h is used as the experimental speed conditions, the preview time is 0.4 s, the integral order and derivative order of the FOPID are both 2, and the vehicle parameters are as shown in Table 1 above.

Figure 13 shows the effect graphs of the trajectory tracking test under U-shaped road conditions with the skilled driver, PID controller, SMC controller, SMC + FOPID controller, and MPC controller.

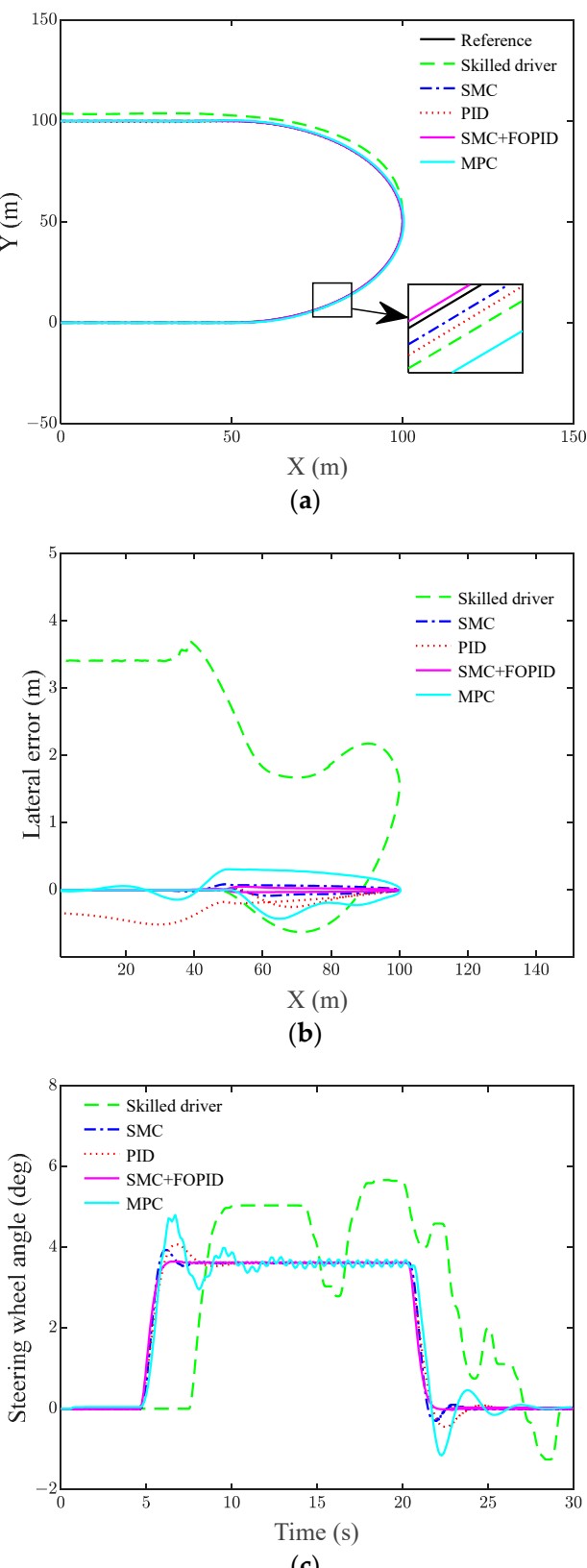

**Figure 13.** *Cont.*

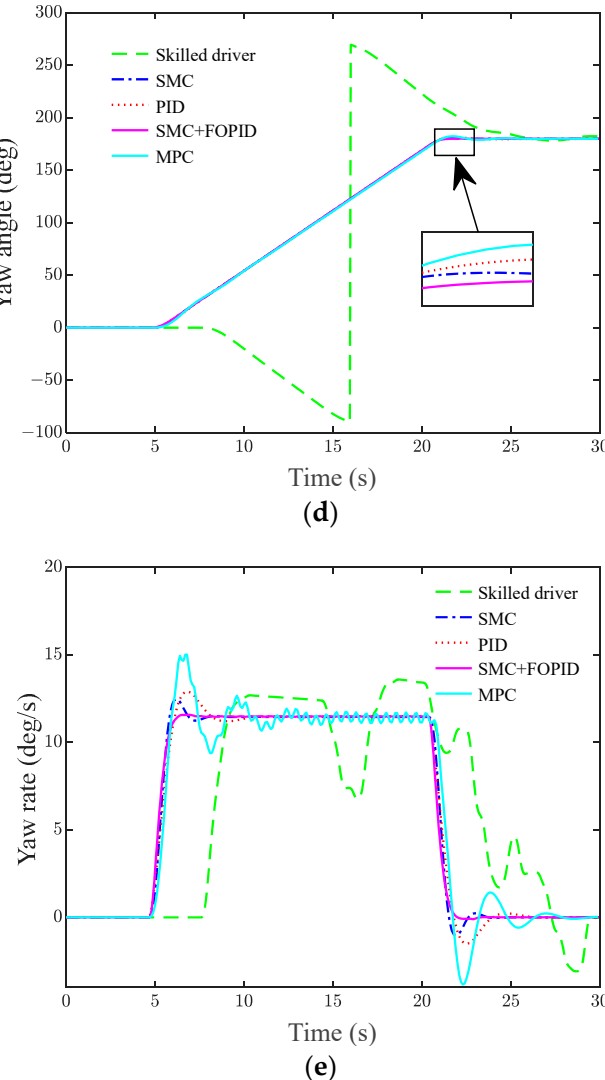

**Figure 13.** Hardware-in-the-loop test results for different control schemes under U-shaped road conditions. (**a**) Lateral position, (**b**) lateral error, (**c**) steering wheel angle, (**d**) yaw angle, and (**e**) yaw rate.

Table 4 was obtained from the data composition of Figure 13.

**Table 4.** Hardware-in-the-loop tracking error under U-shaped road conditions.

| Controller Type | Root Mean Square of Lateral Error (m) | Max Lateral Error (m) |
| --- | --- | --- |
| Skilled drivers | 1.457 | 3.693 |
| SMC | 0.025 | 0.087 |
| PID | 0.154 | 0.512 |
| SMC + FOPID | 0.011 | 0.051 |
| MPC | 0.113 | 0.426 |

As seen in Table 4, the root mean square values of the lateral error for the skilled driver, SMC, PID, MPC, and SMC + FOPID are 1.457 m, 0.025 m, 0.154 m, 0.113 m, and 0.011 m, respectively. The maximum values of the lateral error are 3.693 m, 0.087 m, 0.512 m, 0.426 m, and 0.051 m, respectively. SMC + FOPID has the smallest root mean square value of the lateral error and the smallest maximum value of the lateral error, so the trajectory tracking accuracy is the highest. From Figure 13c,e, it can be seen that the steering disk angle curve and yaw rate curve of SMC + FOPID are the smoothest without abrupt changes.

Therefore, in tracking U-shaped trajectories, the SMC + FOPID controller has a more substantial trajectory tracking capability compared to the other listed controllers.

## 6. Conclusions

The lateral motion control of self-driving vehicle trajectory tracking is studied.

1.  Based on a two-degree-of-freedom vehicle dynamics model and a single-point preview model, combined with the sliding mode control method and the fractional-order proportional-integral-differential control method, the lateral controller of self-driving vehicle trajectory tracking is designed.
2.  In the FOPID controller, the integral and derivative orders can be freely adjusted from 0 to 2, and the lag phase angle for fractional-order integrals and the overtravel phase angle for derivatives are extended from $0°$~$90°$ to $0°$~$180°$. Memory functions for integral and derivative terms allow the system to realize more comprehensive parameter adjustments.
3.  Twelve real drivers are selected to perform directional control for the given road working conditions, and data are collected for comparison tests. Simulation tests are conducted for the four controllers to verify the actual control effect of the SMC + FOPID controller under lane change steering conditions at different speeds.
4.  The final simulation and hardware-in-the-loop test results show that the designed controller can make the self-driving vehicle not only have a high trajectory tracking accuracy, but also ensure the stability of the vehicle in driving.

In this paper, due to the incompleteness of the relevant experimental equipment, the investigation is only carried out up to the hardware-in-the-loop validation session, and does not include the practice session of the specific constraints. In future work, we will continue to study and verify the application of this method in the practice session of the environment and constraints.

**Author Contributions:** Conceptualization, X.Z. (Xiqing Zhang); methodology, X.Z. (Xiqing Zhang) and J.L.; software, X.Z. (Xiqing Zhang) and J.L.; validation, J.L.; formal analysis, Z.M. and D.C.; investigation, X.Z. (Xiaoxu Zhou) and J.L.; resources, X.Z. (Xiqing Zhang); data curation, J.L.; writing—original draft preparation, X.Z. (Xiqing Zhang) and J.L.; writing—review and editing, X.Z. (Xiqing Zhang) and J.L.; visualization, Z.M. and D.C.; supervision, X.Z. (Xiaoxu Zhou) project administration, X.Z. (Xiqing Zhang) and J.L.; funding acquisition, X.Z. (Xiaoxu Zhou). All authors have read and agreed to the published version of the manuscript.

**Funding:** This study was funded by the Shanxi Province Basic Research Program, grant number 202203021221160.

**Data Availability Statement:** Data are contained within the article.

**Conflicts of Interest:** Author Xiaoxu Zhou was employed by the Shanxi Intelligent Transportation Research Institute Company Limited. The remaining authors declare that the research was conducted in the absence of any commercial or financial relationships that could be construed as a potential conflict of interest.

## Nomenclature

| | | | |
|---|---|---|---|
| $\dot{\omega}$ | Derivative of the yaw angle rate | $\dot{\beta}$ | Derivative of the sideslip angle of the vehicle |
| $\omega$ | Yaw rate of the vehicle | $\beta$ | Sideslip angle |
| $l_f$ | Distances from the center of mass to the front axles | $l_r$ | Distances from the center of mass to the rear axles |
| $C_f$ | Vehicle cornering stiffness of the front tire | $C_r$ | Vehicle cornering stiffness of the rear tire |
| $\delta$ | Front wheel angle of the vehicle | $I_z$ | Vehicle rotational inertia around the z axis |
| $v_y$ | Vehicle lateral speed | $v_x$ | Vehicle longitudinal speed |
| $m$ | Mass of the whole vehicle | $R_w$ | Steady-state gain |

| | | | |
|---|---|---|---|
| $L$ | Distance from the front axis to the rear axis | $K$ | Stability factor |
| $\omega_d$ | Ideal yaw rate | $s$ | Form of the sliding mode surface |
| $\eta$ | Gain of the sliding mode controller | $\delta_e$ | Equivalent control quantity |
| $\delta_d$ | Switching control quantity | $e(t)$ | System error |
| $K_p$ | Proportional gains | $K_i$ | Integral gains |
| $K_d$ | Derivative gains | $\chi$ | Integral order |
| $\gamma$ | Derivative order | $\Delta\omega$ | Ideal yaw rate compensation amounts |
| $s^{-\chi}$ | Integral of order with respect to the system error | $s^{\gamma}$ | Derivative of order with respect to the system error |

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
