# Peer review of "Lateral Trajectory Tracking of Self-Driving Vehicles Based on Sliding Mode and Fractional-Order Proportional-Integral-Derivative Control"

_actuators, doi:10.3390/act13010007_

Round 1
Reviewer 1 Report
Comments and Suggestions for Authors
In this paper, a combination of sliding mode control and FOPID is proposed for the application of trajectory tracking of self-driving vehicles. Although the method has been verified through plenty of simulations, following suggestions can be considered.
1. The main contributions are not very clear, more comparative analyses are suggested.
2. Why do the authors choose the combination method of SMC and FOPID? Actually, there are many ways to improve the performance of SMC. What is the main advantage of the proposed method?
3. Although plenty of simulations have been conducted, the simulation conditions are not rich enough, and it is unfair to only compare the proposed method with PID and SMC methods. It is recommended to compare the proposed method with more existing methods in the current literature.
4. Can the method proposed in this article be applied in practice? Are there any limitations?
Comments on the Quality of English Language
The expression of this paper can be further polished.
Reviewer 2 Report
Comments and Suggestions for Authors
In this manuscript, the authors propose a controller based on sliding mode and Fractional-Order control theory. The combination of sliding mode control and Fractional-Order theory is not new. Could you highlight the innovation of NN-MPC?
There are several major concerns regarding the manuscript. It is quite challenging to follow the narrative of much of the central part of the manuscript. Additionally, much of the discussion regarding the results is not very technically rigorous. Finally, specific choices in the writing style and the number of typographical mistakes throughout the manuscript impede the comprehension of the technical content.
So I'm giving this article a rejection
Comments on the Quality of English Languagethis paper require moderate editing of English language
Reviewer 3 Report
Comments and Suggestions for Authors
This paper presents a hybrid control technique for traj. tracking of an autonomous vehicle. In contrast to many of other reported works, the present paper contains HIL (Hardware In Loop) results.
- In which aspect is the model derived in Section 2 (Control System Model for Lateral Motion) different from the ones reported in the literature? - Since there are a lot of symbols, having a Table of nomenclature will help the readers. Include this in the appendix. - Please analyse and report your results quantitatively including in the Abstract. e.g. How much 'good' is good? How much 'better' is better? How much 'smoother' is smoother? - Explicitly mention scientific contribution of this research preferably in a bulleted form list at the end of Section 1 (Introduction) - Discussion on HIL results needs to be more critical and should be presented from a practical real-world perspective. - Page 13, Line 318 include a version of MATLAB/Simulink on which simulations were conducted. Also, mention the specification of the machine on which the proposed control technique was simulated. - Line 68, the literature review on SMC based control methods in various application domains could be updated with DOI: 10.5755/j01.eee.22.1.14094 and "On the derivation of novel model and sophisticated control of flexible joint manipulator" - What are the assumptions made in designing the control law? Mention these clearly and their real-world impact on an actual autonomous vehicle. - At some points et al. has been written in italics while at other points, it is not. Please make them appear consistently. Comments on the Quality of English LanguageThe manuscript requires moderate English changes.
Reviewer 4 Report
Comments and Suggestions for Authors
In this manuscript, the authors present a lateral control strategy based on the FOPID controller and the sliding-mode controller for the self-driving vehicle. The structure is clear. Stability analysis and some hardware-in-the-loop results are provided. Only the following issues need to be dealt with.
1) In the line 15 of the abstract, it should be “…near the sliding mode surface…” rather than “…near the slip mode surface…”.
2) There is no need to present the abbreviations SMC and FOPID in the abstract since they are not used in the following content.
3) The full name of SMC should be given in the introduction section.
4) In the line 61, there is no need to present the abbreviation LQR since it is not used in the following content.
5) In the line 73, it should be “…sliding mode control…sliding surface” rather than “…slide mode control…slide surface”.
Comments on the Quality of English LanguageModerate editing of English language is required.
Round 2
Reviewer 2 Report
Comments and Suggestions for Authors
The paper propose a combination of sliding mode control (SMC) and fractional-order proportional-integral-derivative control (FOPID) for lateral trajectory tracking. They address issues like high-frequency oscillation near the sliding mode surface and modeling errors in the single-point preview model. The paper includes a detailed methodology, simulations, and comparisons with other controllers, demonstrating the improved performance of the proposed SMC+FOPID controller.
Please ensure that all figures and tables are clear and properly labeled. Including more detailed captions could enhance their effectiveness.
Comments on the Quality of English LanguageThe language of the paper appears to be technical and appropriate. However, there are a few areas where language improvements could be beneficial. Some sentences might be overly complex or lengthy. Simplifying these can enhance clarity without compromising the technical content.
Reviewer 3 Report
Comments and Suggestions for Authors
Authors have addressed all of my comments. Please make sure that the resolution of figures in results is adequate for readability.
Comments on the Quality of English LanguageModerate editing of English language required.
